# Current Status and the Epidemiology of Malaria in the Middle East Region and Beyond

**DOI:** 10.3390/microorganisms9020338

**Published:** 2021-02-09

**Authors:** Mohammad Al-Awadhi, Suhail Ahmad, Jamshaid Iqbal

**Affiliations:** Department of Microbiology, Faculty of Medicine, Kuwait University, P.O. Box 24923, Safat 13110, Kuwait; mohammad.alawadhi@grad.ku.edu.kw (M.A.-A.); suhail.ahmad@ku.edu.kw (S.A.)

**Keywords:** malaria, imported malaria, epidemiology, Middle East Region, prevalence

## Abstract

Vector-borne parasitic infectious diseases are important causes of morbidity and mortality globally. Malaria is one of the most common vector-borne parasitic infection and is caused by five *Plasmodium* species, namely *P. falciparum, P. vivax, P. ovale, P. malariae,* and *P. knowlesi*. Epidemiologically, differences in the patterns of malaria cases, causative agent, disease severity, antimicrobial resistance, and mortality exist across diverse geographical regions. The world witnessed 229 million malaria cases which resulted in 409,000 deaths in 2019 alone. Although malaria cases are reported from 87 countries globally, Africa bears the brunt of these infections and deaths as nearly 94% of total malaria cases and deaths occur in this continent, particularly in sub-Saharan Africa. Most of the Middle East Region countries are malaria-free as no indigenous cases of infection have been described in recent years. However, imported cases of malaria continue to occur as some of these countries. Indeed, the six Gulf Cooperation Council (GCC) countries have large expatriate population originating from malaria endemic countries. In this review, the current status and epidemiology of malaria in the Middle East Region countries and other malaria-endemic countries that are home to a large migrant workforce being employed in Middle East Region countries are discussed.

## 1. Introduction

Despite great developments in human healthcare, parasitic infectious diseases still cause significant morbidity and mortality worldwide. It is estimated that the human population experienced more than 2 billion cases of parasitic diseases in 2013 alone [1]. Vector-borne parasitic diseases account for more than 15% of all infectious diseases, causing more than 600,000 deaths annually [2]. Malaria is one of the most common vector-borne parasitic infections which causes more than 400,000 deaths every year globally, most of them among children under five years of age [2].

Three infectious diseases, viz. human immunodeficiency virus (HIV) infection, tuberculosis, and malaria (HTM), received special attention in the year 2000 through the formulation of the millennium development goal (MDG) 6 during the millennium summit at the United Nations. Subsequently, a global fund was created to fight morbidity and mortality due to HTM diseases in 2002. Nearly $11.3 billion were spent by non-governmental foundations/organizations in endemic countries on malaria alone through development assistance for health during 2000 to 2011 [3]. These efforts have resulted in substantial progress as 1.5 billion malaria cases and 7.6 million malaria-related deaths have been averted since the beginning of the new millennium [4]. Despite these advancements, >400,000 people die of malaria every year with nearly two-thirds of the deaths occurring among children <5-years old. Increasing incidence of drug-resistant malaria is also a serious threat to global malaria control efforts.

## 2. Global Epidemiology of Malaria

Even though widespread control and elimination measures were implemented through international and national malaria control programs, malaria continues to be the most important parasitic disease worldwide. The Global Malaria Eradication Program initiated in 1969 ended in failure as hundreds of millions of people were infected with malaria, tens of millions of individuals died (mostly in sub-Saharan Africa), hundreds of thousands of pregnant women died during delivery due to malaria-related complications, and millions of children were born with low birthweight, causing early death or disability [4]. However, the first two decades of the current century in the new millennium represent a golden era in the history of malaria control [4]. According to the recent annual global malaria report published by the World Health Organization (WHO), there were an estimated 229 million malaria cases in 2019 in 87 malaria-endemic countries, a decline of 9 million cases from the year 2000. However, they were higher than the 218 million estimated malaria cases for the baseline year 2015 reported at the Global Technical Strategy (GTS) for malaria 2016–2030 [4]. This is reflected in the decline in the global malaria case incidence (cases per 1000 population at risk) from 80 in 2000 to 58 in 2015 and 57 in 2019. Thus, global malaria case incidence declined by 27% between 2000 and 2015, and by <2% between 2015 and 2019, indicating a slowing of the rate of decline since 2015. Malaria in humans is caused by five *Plasmodium* species, namely *P. falciparum, P. vivax, P. ovale, P. malariae,* and *P. knowlesi*. Most malaria infections in Africa are caused by *P. falciparum* which is also more virulent and causes the majority of malaria-related mortality worldwide. However, increasing prevalence of *P. vivax* infections, particularly in the Indian sub-continent, poses unique diagnostic and therapeutic challenges [5]. Infection with *P. vivax* also results in persistence of the parasite as dormant liver-stage hypnozoites which can cause recurrent episodes of malaria [5]. As a result, *P. vivax* often causes imported malaria cases predominantly among adult men in settings outside sub-Saharan Africa [6]. *P. falciparum* malaria accounted for nearly 99% of cases in Africa and 94% of all malaria cases and deaths globally in 2019 [4].

Although 29 countries accounted for 95% of malaria cases globally, the WHO African Region accounted for 94% (215 million) of cases in 2019 with Nigeria (27%), the Democratic Republic of the Congo (12%), Uganda (5%), Mozambique (4%) and Niger (3%) accounting for ~51% of all cases. However, the malaria case incidence in the WHO African Region was reduced from 363 in 2000 to 225 cases per 1000 population at risk in 2019, largely due to rapid population increase [4]. The WHO South-East Asia Region accounted for ~3% of malaria cases, a 73% reduction from 23 million in 2000 to ~6.3 million in 2019 (India contributing the largest reductions) and a 78% reduction in malaria case incidence, from ~18 cases per 1000 population at risk in 2000 to ~4 cases in 2019 [4]. Malaria cases were also reduced by 26% in the WHO Eastern Mediterranean Region, from ~7 million cases in 2000 to ~5 million (Sudan accounting for ~46% of cases) in 2019. Malaria case incidence also declined from 20 in 2000 to 10 in 2019 [4]. The WHO Western Pacific Region also witnessed a decline of 43% (from 3 million cases in 2000 to 1.7 million cases in 2019) in malaria cases and malaria case incidence (per 1000 population at risk) was reduced from five cases in 2000 to two cases in 2019 [4]. Similarly, malaria cases were reduced by 40% (from 1.5 million to 0.9 million) and case incidence by 57% (from 14 to six) in the WHO Region of the Americas. The WHO European Region has been free of malaria since 2015 [4].

In previous decades, the estimated annual malaria deaths increased from 888,000 in 1990 to a peak of 1.2 million in 2004, after which a 31.5% drop in the number of child deaths in sub-Saharan Africa occurred leading to about 855,000 deaths in 2013 [7]. Since 1990, malaria mortality outside of Africa has been steadily declining. From the year 2000, all regions had decreases in age-standardized incidence and death rates which dropped by 38% in central Asia as a result of increased malaria elimination efforts [7,8]. Globally, malaria caused 409,000 deaths in 2019 with 95% of deaths occurring in 31 countries and 51% of all malaria deaths occurring in only six countries (Nigeria, 23%; Democratic Republic of Congo, 11%; Tanzania, 5%; Mozambique, 4%; Niger, 4% and Burkina Faso, 4%) [4]. About 67% of all malaria deaths occurred in children aged five years or less. Malaria-related deaths have declined steadily during the period 2000–2019, from 736,000 in 2000 to 409,000 in 2019. The malaria mortality rate (i.e., deaths per 100,000 population at risk) has also declined from ~25/100,000 population in 2000 to 12/100,000 in 2015 but only to 10/100,000 in 2019 indicating decreasing rate of decline in later years [4]. The largest reduction in malaria mortality rate (74%) occurred in WHO South-East Asia Region, from ~35,000 in 2000 to 9,000 in 2019 while in absolute numbers, the largest reduction occurred in the WHO African Region, from 680,000 in 2000 to 384,000 in 2019 [4].

Imported malaria cases into non-endemic regions and malaria-free countries are being increasingly recognized as a new public health challenge for the industrialized and other malaria non-endemic countries. Changes in the ecosystem and climate due to global warming have also increased the risk of vector-borne diseases such as malaria [9]. The ever-increasing travel for business and/or leisure and migratory movements for employment or due to geopolitical conflicts have changed the epidemiological characteristics of imported malaria in many malaria-free and non-endemic countries [10,11,12].

Two other major areas of concern were also apparent from the latest global malaria report. Deletions in the *P. falciparum* histidine-rich protein (pfhrp)2 and pfhrp3 genes have been confirmed in 11 (China, Equatorial Guinea, Ethiopia, Ghana, Myanmar, Nigeria, Sudan, Uganda, United Kingdom (imported cases), Tanzania, and Zambia) countries [4]. This renders the parasites undetectable by rapid diagnostic tests based on detection of HRP2. Furthermore, mutations in PfKelch13 have been identified which confer partial resistance to artemisinin, first-line treatment for *P. falciparum* infections [4]. Resistance to other drugs (antifolates, naphthoquinones, antibiotics like clindamycin and doxycycline and 4-aminoquinolines) has also emerged in malarial parasites and few novel targets have recently been identified for the development of new antimalarial drugs [13].

## 3. Epidemiology of Malaria in the Middle East Region Countries

Seventeen countries are located in the Middle East Region which extends to Iran in the East, Egypt in the west, Turkey in the north and Yemen in the south (Figure 1). These countries include Bahrain, Cyprus, Egypt, Iran, Iraq, Israel, Jordan, Kuwait (State of Kuwait), Lebanon, Oman, Palestine (West Bank and Gaza strip), Qatar, Saudi Arabia, Syria, Turkey, United Arab Emirates (UAE), and Yemen. Most of these countries occupy the Arabian Peninsula (Figure 1). They are home to ~5% (corresponding to ~400 million people) of the world’s total population with Egypt, Iran and Turkey being the three most populous countries [4]. Nearly all (except Turkey, Cyprus and Israel which are included in the WHO European region) Middle East Region countries are included in the WHO Eastern Mediterranean Region. The WHO Eastern Mediterranean Region additionally contains 7 other (Afghanistan, Djibouti, Libya, Morocco, Somalia, Sudan and Tunisia) countries in addition to 14 countries included in the Middle East Region [4]. None of the 25 high malaria burden countries are located in the Middle East Region [4]. Indigenous malaria cases have been reported mainly from two countries from the Middle East Region, namely Yemen and Saudi Arabia while the remaining countries have mostly reported imported malaria cases during the last couple of years [4]. The imported malaria cases reported in more recent and selected studies from the Middle East Region countries are presented in Table 1.

The Gulf Cooperation Council (GCC) includes Six Middle East Region countries (Bahrain, Qatar, Kuwait, UAE, Oman and Saudi Arabia) located in the Arabian Peninsula. These affluent countries have many common characteristics such as high income generated from oil wells, modern infrastructure as a result of rapid development and urbanization during the last 30–40 years. These countries employ a large and rapidly changing expatriate population working in both public and private sectors, mostly involving low-wage jobs. This expatriate population outnumbers the citizens in many GCC countries. The expatriates mostly originate from poor/developing countries of south/southeast Asia (e.g., India, Pakistan, Afghanistan, Nepal, Bangladesh, Sri Lanka, and Philippines) and Africa (e.g., Ethiopia, Sudan and Nigeria) where many infectious diseases such as malaria, tuberculosis, taeniasis etc. are endemic [4,5,14,15,16,17,18,19,20]. The number of malaria cases reported from the countries from which most of the expatriates working in GCC countries originate and the number of expatriates from these countries presently living in GCC countries are provided in Table 2. Every year, millions of these expatriates visit their native countries, particularly during summer holidays. Most of the imported malaria cases in GCC countries are detected among the returning expatriate workers after their holidays or new expatriates who come to work. For these reasons, the GCC countries will be discussed as a sub-group in this article.

## 4. Epidemiology of Malaria among GCC Countries

Successful control programs have disrupted local malaria transmission in nearly all GCC countries, and they are now free from indigenous malaria cases with the exception of few sites in Saudi Arabia. However, a continuous influx of imported malaria cases via expatriate workers originating from malaria endemic countries of Africa and south/SouthEast Asia sustains the threat of local transmission of infection [7,14,21]. Nearly all malaria cases in GCC countries occur among expatriates particularly among individuals originating from malaria-endemic countries while a few travel-related infections also occur among GCC nationals. A brief account of the recent developments and the current epidemiology of malaria in GCC countries is described below.

### 4.1. Epidemiology of Malaria in Bahrain

Bahrain is the smallest island country in the Arabian Gulf region with the smallest (~1.6 million) population among GCC countries. Expatriate workers and their dependents comprise nearly 50% of the total population of Bahrain (https://www.mia.gov.bh/kingdom-of-bahrain/population-and-demographics/?lang=en). Malaria was eradicated in Bahrain in the 1980s. However, imported malaria cases continued to occur due to the large immigrant population originating from malaria endemic countries including India, Pakistan, Bangladesh, Philippines and some African countries. Only scant information is available about the current prevalence of imported malaria cases in Bahrain. Only one study carried out at the beginning of the new millennium reported 1572 cases of malaria over a ten-year period (from 1992 to 2001) and all were imported malaria cases (Table 2). The study also reported a consistent decline in the number of malaria cases, from a peak of 282 malaria cases in 1992 to only 54 cases in 2001 [22]. A vast majority (84%) of cases occurred among expatriate population originating from India, Pakistan, Sri Lanka, Bangladesh and Sudan which are malaria-endemic countries. *Plasmodium vivax* accounted for 86% (1346/1572) of cases, while *P. falciparum* infection was only reported in 14% (220/1572) of cases. Only a few breeding sites were identified where *Anopheles* mosquito larvae could be detected, suggesting a low risk of local transmission of infection [22].

### 4.2. Epidemiology of Malaria in Qatar

Qatar is the second smallest GCC country with a total population of 2.8 million inhabitants in 2020 (https://www.onlineqatar.com/visiting/tourist-information/qatar-population-and-expat-nationalities). In 2017, Qatar’s population of 2.6 million comprised 12% Qatari nationals and 88% expatriate workers and their dependents. Several studies have reported on the current situation of malaria in Qatar. An earlier study carried out in the last decade showed that the incidence of malaria which had progressively declined from 58.6 cases/100,000 population in 1997 to 9.5 cases/100,000 population in 2004, started to increase from 2005 onward. The increasing incidence was related with travel of expatriate workers to their native, malaria-endemic countries including India, Pakistan, and Sudan and all patients were imported malaria cases [23]. This trend was confirmed by a subsequent study which also showed an increasing trend in reported cases from 2008 to 2015 and almost all malaria cases were identified among non-Qatari (99.6%) male (93%) expatriate subjects while malaria cases among Qatari nationals were seen among returning travelers [24] (see Table 1 and Table 2).

Another study using PCR-based methods showed that most imported malaria cases in Qatar between 2013 and 2016 were due to *P. vivax*, while *P. falciparum* and *P. falciparum*/*P. vivax* mixed infection cases were less frequent [37]. *P. vivax* infections mostly originated from the Indian subcontinent, while *P. falciparum* infections were mostly presented by African expatriate patients. Furthermore, imported *P. falciparum* strains were highly diverse and a high prevalence of mutations implicated in drug resistance was also observed among malaria parasites. Novel mutations were also detected in Pfkelch13 which confer resistance to artemisinin [37]. Another study has also confirmed this increasing trend in the number of malaria cases in Qatar from 2013 to 2017, with an overall positivity rate of 0.2% among blood donations [38]. A more recent study has shown that 448 patients tested positive for malaria (all imported malaria cases) in Qatar from January 2013 to October 2016. Species-specific PCR identified *P. vivax* malaria in the majority (318 of 448, 71%) of subjects, 118 patients presented with *P. falciparum* malaria while 12 patients had *P. vivax/P. falciparum* mixed infection (Table 1) [25]. High genetic diversity was also seen among imported *P. vivax* parasites. The malaria parasite strains were genotypically comparable to parasites in the Indian subcontinent (mainly India) and East Africa (Sudan and Ethiopia). Furthermore, gametocytes were also detected by a highly sensitive quantitative real-time PCR-based method which represent a major threat for re-introduction of malaria in Qatar. Although indigenous malaria cases were not reported, the authors concluded that the genetic relatedness between *P. vivax* detected in Qatar with those in India will require an elimination strategy targeting flow and dispersal of imported malaria into the region by returning expatriates from endemic countries [25].

### 4.3. Epidemiology of Malaria in Kuwait

The population of Kuwait (~4.5 million) in 2019 included ~30% Kuwaiti nationals and 70% expatriate workers or their family members (https://www.paci.gov.kw/Default.aspx). Most expatriates in Kuwait originate from south/south-east Asian (such as Pakistan, India, Nepal, Bangladesh, Sri Lanka and Philippines) and African (such as Ethiopia, Nigeria and Sudan) countries where many infectious diseases such as malaria, tuberculosis, taeniasis are endemic [4,19,20,39,40,41]. Although Kuwait is a malaria-free country, due to the absence of *Anopheles* mosquito hosts, a scarcity of rain, and lack of freshwater bodies, imported cases of malaria have been regularly reported since the 1980s among migrant workers arriving mainly from malaria-endemic south/south-east Asian and African countries [42,43,44,45,46]. The epidemiological features of imported malaria in Kuwait are similar to other neighboring Gulf countries [14,22,23]. An earlier study conducted during 1985–2000 showed an infection rate of >1200 cases/year and all cases included imported malaria. These cases were either newly arriving expatriate workers from endemic countries or resident workers who were returning to Kuwait after visiting their native countries [42,43,44,45,46]. However, the data from these and a more recent study have shown a declining trend in the incidence of imported malaria cases in Kuwait [26,43,45].

The study carried out in the last decade in Kuwait reported nearly 250–400 malaria cases annually and all cases represented imported malaria cases. A vast majority of cases were detected among expatriates and 80% of cases occurred among male expatriates aged 21–40 years [26]. *P. vivax* was responsible for 71% of malaria cases while *P. falciparum* infections accounted for 27% of malaria cases in Kuwait [26]. A more recent epidemiological study which covered the period from 2013 to 2018 reported that 1913 (25.9%) subjects out of a total of 7386 suspected cases were positive for malaria. The positivity ranged from 262 cases in 2014 to 409 cases in 2017 and a calculated malaria incidence of 6.8/100,000 population in 2014 and 9.9/100,000 population in 2017 [26]. The study further showed that the overwhelming majority of malaria cases (*n* = 1895; 99.1%) were detected in the expatriate population, of which >75% came from Asian countries. The highest number of imported malaria cases (*n* = 1012; 52.9%) were detected among Indian subjects which also form the largest single ethnic group among expatriate residents in Kuwait. Other major malaria-positive groups included expatriate workers from Pakistan (380 subjects) and Afghanistan (94 subjects). Only 18 Kuwaiti nationals tested positive for malaria infection, all with a history of travel to malaria-endemic African countries. An important finding of the study was the detection of mixed *P. falciparum* and *P. vivax* infection in the majority (*n* = 1383; 72.3%) of malaria cases (Table 1) and nearly all the mixed infection cases (97%) were found among subjects from India [26]. Although no autochthonous malaria cases have been detected in Kuwait, certain risk factors for local transmission of infection have recently emerged which include a change in the present ecological and climate conditions due to an enthusiastic drive of making Kuwait green and extending plantations. These conditions may raise larval density levels of *A. stephensi* and *A. pulcherrimus* and consequently support local transmission of infection. Thus, many challenges will need to be overcome to prevent reintroduction and the local transmission of malaria into malaria-free Kuwait [26].

### 4.4. Epidemiology of Malaria in the United Arab Emirates (UAE)

The UAE is the second most populous country among the GCC countries with a total population of nearly 10 million inhabitants in 2020 (https://www.dubai-online.com/essential/uae-population-and-demographics/). The population of 9 million in UAE in 2014 included only 20% Emirati nationals while the remaining 80% inhabitants were expatriate workers and their family members. Only scant information is available on the incidence of malaria in the UAE in recent years. The UAE was certified by the WHO to be free of malaria transmission in 2007, however, imported malaria cases are reported every year among migrant workers or returning travelers from malaria endemic countries [2]. Nilles et al. [27] reported the epidemiological and clinical characteristics of imported malaria in the UAE in 2014. This study, carried out at a single (largest) hospital in Dubai from 2008 to 2010, detected 629 malaria cases which required 162 hospitalizations (eight requiring intensive care support) and 1 death was also reported. A vast majority (493 of 629, 78%) of cases were due to *P. vivax* while only 122 (19%) were caused by *P. falciparum*. Additionally, 14 cases of mixed *P. vivax/P. falciparum* infections were also reported (Table 1). Most (90.1%) of the infected individuals were from India or Pakistan while 7% of the patients were from sub-Saharan Africa [27]. No cases were detected among the local Emirati population. The authors concluded that imported malaria remains an important cause of morbidity in the UAE [27].

### 4.5. Epidemiology of Malaria in Oman

Oman is the second largest country among GCC countries with a total population of 4.5 million inhabitants in 2020 (https://data.gov.om/data/#topic=Population). Omanis accounted for 57.7% of the 4.7 million total population in 2019 while the remaining 42.3% inhabitants were expatriates. Although Oman was a malaria-endemic country until 1990 with approximately 33,000 malaria cases reported every year, malaria eradication measures adopted by the Ministry of Health, Oman (https://www.moh.gov.om) successfully reduced the incidence of malaria to 1 case per 10,000 population in 2000 [28]. Most cases were caused by *P. vivax* and the last documented indigenous malaria case was detected in 2010 [28]. However, a local outbreak of *P. vivax* involving 54 cases over a span of 50 days occurred in 2014 among migrant workers [28]. One patient (Omani national) had previously travelled to Pakistan while the remaining patients were from Bangladesh (*n* = 32), India (*n* = 14), Pakistan (*n* = 6) and Egypt (*n* = 1). However, fingerprinting studies showed that the parasites were not closely related to samples from India, Bangladesh or Pakistan. Although the exact source of this outbreak could not be established, multi-locus sequence data were consistent with the hypothesis of a series of secondary locally acquired cases contracted from a gametocyte carrier infected outside Oman, most likely Iran. Although mosquito larvae were isolated from irrigation pools and large water tanks used in construction sites, this small outbreak of imported malaria cases was not associated with re-emergence of malaria transmission in Oman, as no new cases were reported after the outbreak had ended due to effective treatment of patients including effective removal of hypnozoites and effective vector eradication from the area [28].

### 4.6. Epidemiology of Malaria in Saudi Arabia

Saudi Arabia is the largest and most populous (population of ~35 million inhabitants in 2020) country in the Arabian Peninsula. According to the 2016 data, Saudis accounted for 63.1% of the total population of 31.8 million while expatriate workers and their dependents accounted for the remaining 36.9% of the inhabitants in Saudi Arabia (https://www.stats.gov.sa/en/5305). Saudi Arabia is the only GCC country from which both imported and locally acquired malaria cases have been reported [7,14,47,48,49,50]. According to the WHO global malaria reports, 61 and 38 cases of locally transmitted indigenous malaria cases were reported in 2018 and 2019, respectively [2,4]. Indigenous malaria cases have mainly been reported from the southwestern regions of Saudi Arabia covering Jazan and Aseer provinces [47,50].

Significant progress has been made in the fight against malaria in Saudi Arabia in the past two decades. The number of autochthonous malaria cases have declined sharply by 99.8% between 1998 and 2012 [14,47,48]. The steep decline in malaria cases was brought about by rapid scaling up of vector control measures, adoption of artesunate and sulfadoxine-pyrimethamine combination treatment and a regional partnership initiated in 2007 for a malaria-free Arabian Peninsula [47]. The draft Global Technical Strategy for malaria adopted in 2002 aimed to eliminate malaria from at least 10 countries by 2020 and Saudi Arabia is one of two (Yemen being the other) countries in the Arabian Peninsula yet to achieve elimination [47,48]. In 2004, a revised action plan was developed to eliminate malaria from Saudi Arabia by preventing re-introduction of malaria into regions which had previously been declared malaria-free, by removing foci of active transmission in the Mecca and Medina areas and a heightened effort for surveillance and control of mosquito breeding foci, to eliminate malaria from the Jazan and Aseer provinces [47,48,49,50].

Several studies have reported on the situation of malaria in Saudi Arabia in recent years. One study reported a declining trend in malaria cases between the year 2000 and 2014 in Jazan Province, where 9936 (64%) cases of imported malaria and 5522 (36%) locally acquired cases (were detected, with an average annual incidence of 0.03 cases per 1000 population [48]. Although indigenous malaria cases declined from 2756 cases in 2000 to 126 cases in 2004 which declined further to 15 cases in 2014, the number of imported malaria cases increased during the same period with nearly 250 to 830 cases detected every year except in the years 2007 and 2009 when 1705 and 1310 imported malaria cases, respectively, were detected [48]. The identification of *Plasmodium* spp. by PCR in another study detected the dominance of *P. falciparum* in Saudi Arabia. More than 98% of 371 malaria-positive blood specimens showed *P. falciparum* infection, while only 7 had *P. falciparum/P. vivax* mixed infection and none had *P. vivax* mono-infection [51]. Another study which surveyed 1840 individuals from 54 villages in Aseer Province during the year 2006–2007 using passive case detection survey found that 49 (2.7%) individuals had malaria; one individual had *P. vivax* and the rest had *P. falciparum* infection [52]. Out of the total malaria cases, 18 (36.7%) were autochthonous while about two-thirds were imported from malaria-endemic countries [52]. Another recent study from Jazan, a low-transmission district in southwestern Saudi Arabia, detected malaria in 30 patients from August 2016 to September 2018 using a species-specific nested PCR method. The authors reported that 80% of cases were imported, 76.6% of cases had *P. falciparum* infection, 16.6% had *P. vivax* infection, and only 6.6% had mixed *P. falciparum/P. vivax* infection [53]. Of 2711 microscopy confirmed cases in 2018, 61 patients were indigenous malaria cases (*P. falciparum*, *n* = 57 and *P. vivax*, *n* = 4) (Table 1) while the remaining 2650 patients represented imported malaria cases [2,4].

## 5. Epidemiology of Malaria in Other Countries in the Middle East Region

### 5.1. Epidemiology of Malaria in Yemen

Yemen with a total population of ~30 million in 2020 is the southernmost country in the Arabian Peninsula. Yemen is also endemic for malaria accounting for the second highest number of projected malaria cases after the leading contributor Sudan, in the WHO Eastern Mediterranean Region [4]. Although substantial reductions in malaria burden were achieved since the launch of the National Malaria Control Program in 2000, malaria control has become challenging due to the displacement of people as a consequence of the current civil war-like situation and humanitarian crisis in recent years. This is reflected from the recent data on the current situation of malaria in the country. Although the number of microscopy-confirmed malaria cases progressively declined from 78,269 in 2010 to 42,052 in 2015, they increased progressively after 2015 to reach 64,233 cases in 2018 [2]. Similarly, the number of all suspected malaria cases also declined from 835,018 in 2010 to 711,680 in 2015 and then increased to 713,908 in 2018 [2]. The number of malaria deaths also increased from 1309 in 2015 to 2138 in 2018 [2]. Nearly all malaria cases in Yemen in recent years are due to *P. falciparum* as 112,823 cases were reported due to this species in 2018 while only 970 cases due to *P. vivax* were detected [2]. Approximately two-thirds of the population are at risk of infection. For this reason, the National Malaria Control Program started using the Integrated Malaria Surveillance System and the Early Disease Electronic Warning System, both of which have been reported to be useful. The former excels in assessing the burden of malaria, response to outbreak, and future planning and the latter is efficient in detecting malaria outbreaks [54].

Hodeidah, a coastal governorate in Tihama Region in the west of Yemen, is the most malaria-afflicted governorate in the country. In a recent study, Alwajeeh et al. [55] screened 400 asymptomatic school children in Bajil district of Hodeidah Governorate and detected *P. falciparum* among 32 of 400 (8.0%) schoolchildren with most infections showing low-level parasitemia. *Plasmodium vivax* was detected in only one child. A major factor in the endemicity of malaria in Yemen is the insecticide resistance in the main malaria vector, *Anopheles arabiensis* [56]. Recently, universal coverage of the targeted malaria-endemic areas with long-lasting insecticidal nets has been implemented as one of the key interventions for malaria control and elimination in Yemen [57].

### 5.2. Epidemiology of Malaria in Iraq

Iraq has been affected by war/civil unrest for about two decades. According to the United Nations (http://www.unhcr.org/syria-emergency.html), there are nearly 0.25 million Syrian refugees in addition to nearly 3 million Iraqi internally displaced persons. Although Iraq is rich in agricultural land and vegetation, freshwater bodies (Euphrates river and Tigris river), and an abundance of *Anopheles* mosquitos, the WHO had reported no cases of malaria in Iraq for three consecutive years since 2011 [2]. Also, no malaria cases or deaths were reported in 2013 [7]. A malaria epidemic occurred in Babylon governorate, Iraq in 1997–1998, but malaria transmission in the area was successfully interrupted subsequently [58]. Although the parasitological survey conducted in 2002 identified no malaria cases in the same area, *Anopheles stephensi* and *A. pulcherrimus* mosquito vectors were detected in high densities. However, no parasite sporozoites or oocysts were detected. It was suggested that malaria transmission could recur if *A. stephensi* indoor resting density exceeds the critical threshold and imported malaria cases are not monitored in Iraq [58]. The last documented case of malaria in Iraq was detected in 2009 [4]. However, there is not much information regarding the current malaria situation in Iraq.

### 5.3. Epidemiology of Malaria in Syria

From the year 1990 to 2000, the annualized rate of change in malaria incidence and deaths in Syria declined by approximately 6.6–6.8% [7]. Since 2004, no autochthonous cases of malaria have been reported [4]. The number of imported cases of malaria had increased from 12 cases in 2002 to 48 cases in 2011 [7]. Little information is available from recent independent studies as the country is in a civil war since 2012. Nonetheless, no malaria cases or deaths occurred from 2000 to 2013 in Syria [4,7].

### 5.4. Epidemiology of Malaria in Jordan

Only scant information is available regarding cases of malaria in Jordan. Jordan is a malaria-free country in the Middle East Region as no indigenous malaria cases have been reported for several years [2,4,7]. However, cases of imported malaria which is a major health concern for countries considered as malaria-free have been reported in two studies. An earlier study described 511 cases of imported malaria among civilian Jordanians returning from Asian and African countries during 1991–2011 [59]. A more recent study showed that 304 imported malaria cases were detected between January 2007 and November 2011. Of these, 192 cases were among Jordanians (mostly military personnel who participated in Peace Keeping Forces with the United Nations) returning home and the remaining 112 cases were detected among foreign nationals who arrived in the country for work or tourism [29]. Most imported cases were detected among patients originating mainly from Eritrea, Côte d’Ivoire, India, Sudan, Liberia, and Pakistan. These infections were due to *P. falciparum* (*n* = 199 cases), *P. vivax* (*n* = 93 cases) and *P. malariae* (*n* = 8 cases). Mixed infection was also detected in 4 cases [29]. Continuous vigilance by health authorities has been emphasized by the authors to avoid reintroduction of the disease into the Jordan.

### 5.5. Epidemiology of Malaria in Lebanon

Information regarding malaria in Lebanon is extremely scare. Lebanon is a malaria-free country in the Middle East Region as no indigenous malaria cases have been reported for several years [2,4,7]. Local transmission of malaria has been eradicated for several decades even though *Anopheles* spp. are present in Lebanon. However, cases of imported malaria have been reported due to the large number of Lebanese nationals in African countries, and their frequent travel to Lebanon. Total confirmed imported malaria cases increased from 55 in 2003 to 115 in 2012. Among imported malaria cases in 2013 and 2014, the majority (>50%) were due to *P. falciparum* infection [30].

### 5.6. Epidemiology of Malaria in Palestine (West Bank and Gaza)

Palestine has a population of about 5 million people distributed in the West Bank and Gaza Strip. Very limited information is available regarding malaria cases in Palestine which has been a malaria-free country since 1925 [4,7,32]. Only seven malaria cases were reported from 2008 to 2017 and all cases represented imported malaria cases as the patients had travelled to malaria-endemic countries before their infection [31].

### 5.7. Epidemiology of Malaria in Israel

Israel is also a malaria-free country in the Middle East Region as no indigenous malaria cases have been reported for several decades [2,4,7]. However, cases of imported malaria have been reported among returning travelers. One study reported 18 cases of *P. malariae* infection among Israeli travelers who had visited Africa from January 2008 to January 2017 [60]. A more recent study carried out at Chaim Sheba Medical Center in Tel Aviv reported that 145 malaria cases were detected among 722 patients hospitalized during 2004 to 2015. All cases occurred among patients after returning to Israel from Asia or Africa. Most (86 of 145, 59%) hospitalized patients tested positive for *P. falciparum* malaria (Table 1) [32].

### 5.8. Epidemiology of Malaria in Iran

According to the World Malaria Report 2020 and other studies on the global epidemiology of malaria; Iran is currently in the malaria elimination phase as no malaria cases have been reported for the last two (2018 and 2019) consecutive years [4]. Iran had previously reported indigenous malaria cases [2,7]. Consistent with recent data, previous studies had also reported a declining trend in the incidence of indigenous malaria cases in Iran [33,61,62]. However, imported malaria cases continue to occur.

Until recently, Iran was considered as one of the malaria endemic countries of the WHO Eastern Mediterranean Region as it neighbors other malaria endemic countries such as Afghanistan and Pakistan. Salmanzadeh et al. [61] reported on the trend of malaria distribution during 2001–2014 in Khuzestan Province, southwestern Iran. A total of 541 malaria confirmed cases were detected which progressively decreased over time. The highest number of infections (*n* = 161) were seen in 2001 and the lowest (*n* = 0) were detected in 2014. Most (508 of 541, 94%) infections were imported malaria cases. *Plasmodium vivax* was the dominant species, identified in 478 (88%) individuals while 63 (12%) patients were infected with *P. falciparum* [61]. Delam et al. [62] described the epidemiological status of malaria in the southern region of Fars Province in southern Iran during 2006–2018. The authors found a total of 190 cases of malaria with >95% infections affecting Afghans and most of them were workers. Azizi et al. [33] reported on malaria trend in East Azerbaijan Province and compared the data with countrywide cases during 2001–2018. A total of 135 malaria cases were detected in East Azerbaijan Province while 154,560 cases were reported throughout the country during 2001 to 2018. The authors reported that the incidence rate decreased in East Azerbaijan Province from 0.4/10,000 in 2001 to zero in 2018 and no indigenous transmission was detected during the last 14 years. The authors also reported that there was a total of 572 malaria cases throughout Iran in 2018 and all were imported malaria cases. There has been no indigenous transmission of malaria in Iran in 2018 and 2019 [4,33].

### 5.9. Epidemiology of Malaria in Turkey

Turkey, a part of the WHO European Region is a malaria-free country. The WHO European Region has been malaria free since 2015 and the last country to report an indigenous malaria case was Tajikistan in 2014 [4]. The malaria incidence declined rapidly during 1990–2013 [7]. Turkey has reported no case of autochthonous malaria since the year 2010, and no deaths since 2000. However, an annual average of 216 imported cases of malaria from 2010 to 2017 were reported (4). Besli et al. [63] described the detection of 42 malaria cases in Istanbul Region between 2002 and 2017. Nearly all malaria infections were imported cases due to *P. vivax* (*n* = 22) or *P. falciparum* (*n* = 19) with 32 of 42 cases infected during their journeys to Africa. In another study, Özkeklikçi & Avcıoğlu [64] identified all malaria cases between January 2005 and December 2015 in Gaziantep, western Turkey. A total of 31 malaria cases were detected including 5 indigenous cases, 2 relapse cases while 24 were imported cases. The last indigenous case of malaria was detected in 2005 while in later years, cases with relapse and cases originating from abroad were detected due to increased travel related to tourism and commercial activities [64]. In another study conducted during 2012–2017 in Antalya Province, 36 malaria cases were detected, all of which were imported malaria cases from abroad [34]. The malarial parasite responsible for these infections was identified as *P. falciparum* (29 patients), *P. vivax* (4 patients), *P. ovale* (2 patients) and *P. malariae* (1 patient) (Table 1). The author concluded that, for Turkey to remain malaria-free, early diagnosis and treatment of the disease and integrated mosquito control programs need to be uninterruptedly maintained [34].

### 5.10. Epidemiology of Malaria in Cyprus

Cyprus, a part of the WHO European Region, is a malaria-free country. Only scant information is available for Cyprus. Although Cyprus was declared malaria free in 1967 and the WHO European region was also declared malaria-free in 2015 for the first time as there were no indigenous cases, imported malaria cases have occasionally been reported. Emms et al. [65] described two cases of *P. vivax* malaria in returning travelers. Another case of *P. vivax* malaria was described by the Centers for Disease Control and Prevention, Cyprus in another returning traveler in 2017 [65]. Imported malaria cases (*n* = 13) have also been reported in returning travelers in Northern Cyprus during 2016–2019 in a more recent study [35]. *P. falciparum* (*n* = 10)*, P. vivax* (*n* = 2) and *P. ovale* (*n* = 1) infection were detected in 13 patients with nine cases detected in 2019 alone [35]. The authors concluded that the rising number of recently imported cases, particularly in foreign patients who travel from the malaria-endemic region, increase the risk of re-emergence of malaria in Cyprus [35].

### 5.11. Epidemiology of Malaria in Egypt

Malaria was endemic in Egypt until the end of the last century, but the prevalence declined steadily and rapidly during the 1990s. Egypt was declared malaria free in 1998 and entered the WHO’s prevention-of-reintroduction phase after sustaining at least three years of no autochthonous malaria transmission [7,14]. However, sporadic cases continued to occur, particularly in the Al-Fayoum Governorate in the northwestern part of Egypt due to a unique combination of hydrogeology, soil variables and the presence of a highly efficient mosquito vector [66]. Malaria cases were also detected in Aswan Governorate, Egypt involving 21 confirmed cases during May–June 2014 and all cases were male subjects with history of travel to malaria-endemic Sudan [36]. No indigenous malaria cases have been detected during the last several years and no malaria-related deaths have been recorded in Egypt during the period 2010–2019 [2,4].

## 6. Epidemiological Data from Countries of Origin for Imported Malaria Cases in GCC and Other Middle East Region Countries

As stated above, many countries in the Middle East Region, particularly GCC countries have a large expatriate population [14,16,17,18,19,39,40,41,42]. Many expatriates originate from malaria endemic countries such as Sudan, Ethiopia and Nigeria in Africa and India, Pakistan, Afghanistan, Bangladesh and Philippines in Asia. Nearly all of these countries are highly endemic for malaria (Table 2). Since the expatriates contribute to a significant number of imported malaria cases in the GCC and some other nearby countries, as described above, a brief account of the current epidemiology of some of these important countries is also briefly given here.

### 6.1. Epidemiology of Malaria in Sudan

Sudan, with nearly 2.4 million malaria cases in 2019, is the leading contributor for malaria in the WHO Eastern Mediterranean Region [4]. Previous studies carried out in several Middle East Region countries have shown that travelers or returning travelers from Sudan have contributed to the number of imported malaria cases in Bahrain [22], Qatar [23,25,37], Kuwait [26,43,45], UAE [27], Saudi Arabia [14,47,49], Jordan [29,59], and Egypt [36]. Furthermore, development of drug resistance in the malarial parasite is a major threat to malaria control programs in Sudan. Some surveys have reported failure of artemisinin-based combination therapies for the treatment of uncomplicated malaria in Sudan and high prevalence of mutations in *P. falciparum* drug resistance genes were detected [67]. In another recent study, high frequency of mutations in *Pfcrt*, *Pfdhfr,* and *Pfdhps* was also detected, which are associated with chloroquine and sulfadoxine-pyrimethamine (SP) resistance in *P. falciparum* [68].

### 6.2. Epidemiology of Malaria in Ehiopia

Ethiopia, with nearly 2.5 million malaria cases in 2019, is the 23rd most common contributor for total malaria cases in the WHO African Region [4]. Previous studies have shown that travelers or returning travelers from Ethiopia have also contributed to imported malaria cases in Qatar [25] and Kuwait [26,43,45]. Although the number of total malaria cases in Ethiopia is still very high, the country has shown notable progress in reducing the burden of malaria over the past two decades and this has shifted efforts from control to elimination of malaria in some regions of the country such as the Harari Region in Eastern Ethiopia [69]. The data showed that malaria incidence in the Harari Region declined from 42.9 cases per 1000 population in 2013 to only 6.7 cases per 1000 population in 2019 and malaria-related deaths decreased from 4.7 deaths per 1,000,000 population annually in 2013 to zero in 2015. Also, *P. falciparum, P. vivax* and mixed infections accounted for 69.2%, 30.6%, and 0.2% of the cases, respectively [69]. Ethiopia has set a goal to eliminate malaria by 2030 and artemether-lumefantrine treatment is being used as one of the cornerstone strategies for uncomplicated *P. falciparum* malaria [70].

### 6.3. Epidemiology of Malaria in Nigeria

With 208 million individuals, Nigeria is the most populous country in Africa. With nearly 60 million malaria cases, Nigeria contributed 27% of all malaria cases and 23% of all malaria deaths globally, the most for any country even in the WHO African Region [4]. Both, the number of malaria cases and malaria deaths have increased in 2019 compared to 2018 which is opposite of what was normally seen in most of the high malaria burden countries [4]. Previous studies have shown that travelers or returning travelers from Nigeria have contributed to imported malaria cases in Qatar [37], Kuwait [26,43,45], Saudi Arabia [14,71,72] and Turkey [63]. Nigeria is also one of 11 countries which have reported Pfhrp2/3 deletions that compromise the utility of HPR2-based diagnostic tests for the detection of malaria cases [4]. Although the levels of *P. falciparum* resistance to sulfadoxine-pyrimethamine have remained low from 2000 to 2020 in western Africa including Nigeria, the rising incidence of drug resistance polymorphisms to first line drugs (artemisinin combination therapy) is a major challenge to control malaria [73,74]

### 6.4. Epidemiology of Malaria in Pakistan

Pakistan is a highly malaria-endemic country with more than 670,000 cases of malaria and 3159 deaths in 2013 [7]. The latest WHO report projected nearly 700,000 malaria cases in Pakistan in 2019 [4]. A recent study in the highly endemic Bannu District in Pakistan examined blood smears of 2033 individuals suspected of malaria infection and reported 429 (21.1%) malaria-positive cases by at least one diagnostic test (microscopy, rapid diagnostic test, RDT or PCR) [75]. The positivity by PCR, microscopy and RDT was 30.5%, 17.7% and 16.4%, respectively. Nearly 80% of malaria cases had *P. vivax* infection, while *P. falciparum* only accounted for 11% of cases and *Plasmodium* spp. mixed infection was seen in 9% of malaria cases [75]. Another study examined blood specimens from 216 febrile patients in a tribe-governed region in Pakistan which receives refugees due to military conflicts from neighboring Afghanistan. The authors reported that 86.5% of the malaria cases were attributed to *P*. *vivax* infection, 11.8% to *P*. *falciparum*, and showed that the ratio of *P. vivax* to *P. falciparum* infection has increased in provinces which border Afghanistan [76]. A recent study which used RDTs to screen 31,041 individuals for malaria infection in three highly endemic districts of Khyber Pakhtunkhwa Province (Bannu, Dera Ismail Khan and Lakki Marwat) reported an overall malaria prevalence of 13.8% in the general population. The species prevalence of *P. vivax*, *P. falciparum,* and *Plasmodium* spp. mixed infection were reported as 92.4%, 4.7%, and 2.9%, respectively [77]. These reports show an increasing dominance of *P. vivax* in highly endemic regions of Pakistan. Previous studies carried out in several Middle East Region countries have shown that travelers or returning travelers from Pakistan have contributed to imported malaria cases in Bahrain [22], Qatar [23,25,37], Kuwait [26,43,45], UAE [27], Saudi Arabia [14,52,71,72], Jordan [29,59], and Egypt [36].

### 6.5. Epidemiology of Malaria in Afghanistan

Afghanistan is another highly malaria-endemic country, where approximately half of the population is at risk of infection [78]. The incidence of malaria in Afghanistan exceeded 200,000 cases in 2013, with 1783 deaths [7]. The latest WHO report projected nearly 400,000 malaria cases in Afghanistan in 2019 [4]. A study of malaria prevalence in Jalalabad City, employing real-time PCR with high resolution melting analysis on 296 blood samples reported asymptomatic malaria in 26 (7.8%) individuals [78]. Most of the imported malaria cases in Iran in one study originated from Afghanistan [62].

### 6.6. Epidemiology of Malaria in India

Although India has contributed to the largest absolute reductions in malaria cases in WHO South-East Asia Region from 2000 to 2019, the latest WHO report still projected nearly 5.6 million malaria cases in India in 2019 [4]. Outbreaks peak during summer and fall monsoon seasons which raise the number of malaria cases and deaths in the country [79,80]. Approximately 95% of the population in India lives in endemic regions where 80% of malaria cases occur in about 20% of communities which inhabit remote, hilly and difficult-to-access regions [2].

One study which employed a PCR assay reported that the overall malaria prevalence in India was 19% which ranged from 6% in Oddanchatram, South India to 35% in Ratnagiri, West India [81]. The study also emphasized that microscopy and rapid diagnostic tests had lower sensitivity than PCR, resulting in underdiagnosed malaria in the rural regions of India [81]. In southwestern regions of India, *P. vivax* infections account for approximately 80% of malaria which have been reported to cause severe malaria leading to more deaths than *P. falciparum* [82]. In Mangaluru City in southwestern India, a study examined a total of 579 malaria patients and reported that 364 (62.9%) had *P. vivax* infection, 150 (25.9%) had *P. falciparum* while 65 (11.2%) patients had *Plasmodium* spp. mixed infection [83]. The majority (506 or 87%) of malaria patients had mild malaria, which may be attributed to prompt treatment or previous exposure to the parasite [83]. Although *P. vivax* was thought to dominate India in comparison to *P. falciparum*, a study which examined 2333 blood samples using microscopy, rapid diagnostic tests and PCR assay collected from 9 malaria-endemic Indian states had reported that the ratio of *P. vivax* to *P. falciparum* infection was 51:49 while 13% of cases had *Plasmodium* spp. mixed infection [84]. Mixed infections have been reported frequently from India [83,84,85]. *P. falciparum* which had dominated India’s malaria cases previously is now showing a decreasing trend over the past few years from 65.4% in 2017 to 46.4% in 2019. The majority of malaria cases in some regions in India are now caused by *P. vivax*. On the contrary, the rising number of malaria cases in some other areas are caused mostly by *P. falciparum* with a minor contribution from *P. vivax*. Nonetheless, India alone is estimated to harbor 47% of *P. vivax* malaria cases worldwide, where 7 out of 36 Indian states (primarily northern/eastern states of Uttar Pradesh, Jharkhand, Chhattisgarh, West Bengal, Gujarat, Madhya Pradesh, and Odisha) account for 90% of malaria cases [2]. Previous studies carried out in several Middle East Region countries have shown that travelers or returning travelers from India have contributed greatly to imported malaria cases in Bahrain [22], Qatar [23,24,25,37], Kuwait [26,42,43,44,45], UAE [27], and Saudi Arabia [14,52,71,72]

### 6.7. Epidemiology of Malaria in Bangladesh

With a population of more than 160 million people, Bangladesh is also a malaria-endemic country where 17.5 million people at risk of malaria infection [4,86]. Nonetheless, Bangladesh has been successfully accelerating its efforts to eliminate malaria from the country and has seen a steady decline in the number of malaria cases during the period 2000–2019. The country was projected to have nearly 50,000 malaria cases in 2019 [4]. The share of *P. vivax* malaria and anti-malarial drug resistance has increased over the past two decades [4]. A meta-analysis of patient data from three sites in Bangladesh showed that 12–26% of *P. falciparum* malaria patients were at risk of *P. vivax* parasitemia after several weeks following treatment [87].

### 6.8. Epidemiology of Malaria in the Philippines

The Philippines is a malaria-endemic country in the WHO Western Pacific Region with more than 550,000 malaria cases and 230 deaths reported in the year 2013 [7]. However, from the year 2010 to 2018, the number of malaria cases and deaths have declined considerably. The regions with highest prevalence of malaria are Palawan Province and Mindanao Islands [2,4]. The country was projected to have nearly 40,000 malaria cases in 2019 [4]. Recent studies have shown mixed *P. falciparum/P. vivax* malaria infections as well as *P. knowlesi* as an important cause of human malaria within Palawan Province in the Philippines [88,89]. Mutations in dihydropteroate synthase (pvdhps) and dihydrofolate reductase (pvdhfr), genes associated with sulfadoxine-pyrimethamine drug resistance, have also been recently reported among *P. vivax* isolates collected in Palawan, in the Philippines [90].

## 7. Conclusions

The highest number of malaria cases and deaths occur in sub-Saharan African and south/southeast Asian countries. However, imported cases of malaria also occur in the Middle East Region and other developed countries which are either considered malaria-free or have eradicated malaria through concerted efforts and have been declared malaria-free recently. Many Middle East Region countries, particularly GCC countries, have large expatriate populations originating from malaria-endemic countries of south/southeast Asia, such as India, Pakistan, Afghanistan, Bangladesh, and the Philippines and Africa, such as Nigeria, Sudan, and Ethiopia. Imported malaria cases into non-endemic regions and malaria-free countries are being increasingly recognized as a new public health challenge. Imported malaria cases from South Asia where *P. vivax* infections are also common, often result in persistence of the parasite as dormant liver-stage hypnozoites which can cause recurrent episodes of malaria in non-endemic settings. Imported *P. vivax* infections can thus result in outbreaks of infection if mosquito vector breeding is facilitated and maintained by artificial water reservoirs such as construction sites as was demonstrated in the outbreak reported from Oman in 2014 which had been malaria-free since 2010.

Changes in the ecosystem and climate due to global warming, the ever-increasing travel for business and/or leisure, migratory movements for employment or due to geopolitical conflicts and plantation drives in previously barren/arid surroundings such as GCC countries have also changed the epidemiological characteristics of imported malaria in many Middle East Region countries. These conditions, if not appropriately managed, may also raise vector density levels to the point that may support local transmission of infection following imported malaria cases in these settings. Increasing incidence of drug-resistant malaria, particularly due to drug-resistant *P. vivax* strains, is also a serious threat to malaria control efforts in malaria-free countries in the Middle East which employ large number of expatriates from *P. vivax*-endemic South Asian countries. Hence, imported cases of malaria should be regularly monitored in such settings. Efficient and more effective vector control measures will also have to be adopted to prevent imported malaria cases leading to indigenous malaria in Middle East Region countries. Thus, for many of the Middle East Region countries to remain malaria free, early diagnosis, appropriate treatment of the disease, integrated mosquito control programs, and regular monitoring for drug resistance in *Plasmodium* spp. need to be uninterruptedly maintained.

## Figures and Tables

**Figure 1 microorganisms-09-00338-f001:**
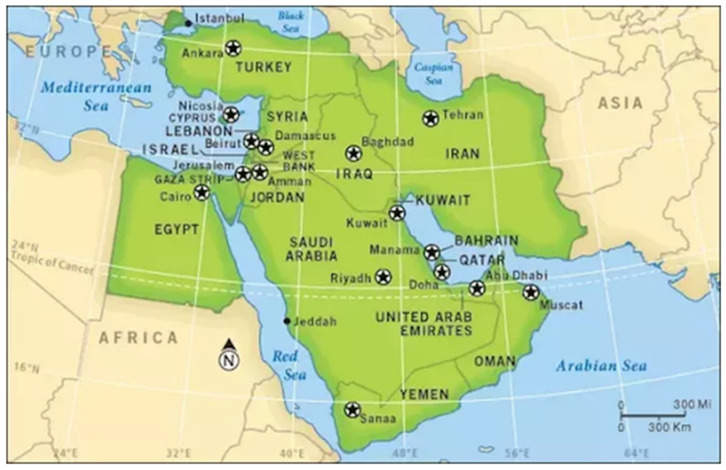
Geographical map showing the location of countries included in the Middle East Region. Reproduced with permission from Associated Marketing Ltd.

**Table 1 microorganisms-09-00338-t001:** Imported and indigenous malaria cases and species distribution reported in selected studies from the Middle East Region countries.

Country	Duration of Study	No. of Malaria Cases Detected			Malaria Infection Caused by				References
		Total	Imported	Indigenous	*P. falciparum*	*P. vivax*	*Pf/Pv* mixed ^d^	Others	
Bahrain ^a^	1992–2001	1572	1572	0	220	1346	5	1	Ismaeel et al., 2004 [22]
Qatar ^a^	2013–2016	448	448	0	118	318	12	0	Abdelraheem et al., 2018 [25]
Kuwait ^a^	2013–2018	1913	1913	0	361	124	1383	45	Iqbal et al., 2020 [26]
UAE ^a^	2008–2010	629	629	0	122	493	14	0	Nilles et al., 2014 [27]
Oman	2014	54	1	53	0	54	0	0	Simon et al., 2017 [28]
Saudi Arabia	2018	2711 ^b^	2650	61	57 ^b^	4 ^b^	0 ^b^	0 ^b^	WHO Malaria Reports [2,4]
Yemen	2018	713,908 ^c^	NA	NA	112,823 ^c^	970 ^c^	63 ^c^	69 ^c^	WHO Malaria Reports [2,4]
Iraq ^a^	NA	NA	NA	NA	NA	NA	NA	NA	WHO Malaria Reports [2,4]
Syria ^a^	NA	NA	NA	NA	NA	NA	NA	NA	WHO Malaria Reports [2,4]
Jordan ^a^	2007–2011	304	304	0	199	93	4	8	Jamain et al., 2013 [29]
Lebanon ^a^	2012	117	117	0	63	NA	NA	NA	WHO Health profile 2015 [30]
Palestine ^a^	2008–2017	7	7	0	NA	NA	NA	NA	Hamarsheh & Amro 2020 [31]
Israel ^a^	2004–2015	145	145	0	86	36	0	23	Avni et al., 2018 [32]
Iran	2018	572	572	0	NA	NA	NA	NA	Azizi et al., 2020 [33]
Turkey ^a^	2012–2017	36	36	0	29	4	0	3	Ser et al., 2020 [34]
Cyprus ^a^	2016–2019	13	13	0	10	2	0	1	Guler et al., 2020 [35]
Egypt ^a^	2014	9	9	0	3	6	0	0	Dahesh et al., 2015 [36]

^a^ No indigenous malaria cases were detected during 2010–2019 according to recent WHO global malaria reports [2,4]. ^b^ Only microscopy confirmed cases are shown and species distribution is shown only for indigenous malaria cases [2]. ^c^ All suspected cases are included among total cases and species distribution is shown for only confirmed cases [2]. ^d^
*Pf/Pv* mixed; *P. falciparum/P. vivax* mixed infections; NA, not available.

**Table 2 microorganisms-09-00338-t002:** No. of malaria cases reported by WHO in 2019 from some malaria endemic countries whose nationals predominate as expatriate workers in GCC countries.

Country	Population (In Millions)	No. of Malaria Cases in 2019 (In Millions)	Population of Expatriates from Indicated Countries among GCC Countries *					
			Bahrain	Qatar	Kuwait	UAE	Oman	Saudi Arabia
India	1385	5.6	318,547	698,088	1,124,256	3,419,875	1,325,444	2,440,489
Pakistan	223	0.7	78,638	235,876	330,824	981,536	240,965	1,447,071
Afghanistan	39	0.4	690	1602	2826	8071	NA	469,324
Bangladesh	165	0.05	82,518	263,086	370,844	1,079,013	304,917	1,246,052
Philippines	110	0.04	50,585	168,461	192,143	556,407	44,546	628,894
Nigeria	208	60	2054	4152	4702	15,465	NA	NA
Sudan	44	2.4	7917	23,954	48,204	131,254	19,155	469,324
Ethiopia	116	2.5	713	1700	3806	10,886	NA	160,192

* Population figures (at mid-year 2019) retrieved from: https://www.un.org/development/desa/pd/content/international-migrant-stock-2019; GCC, Gulf Cooperation Council; UAE, United Arab Emirates; NA, not available.

## Data Availability

Data sharing not applicable.

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
