# Peer review of "Current Status and the Epidemiology of Malaria in the Middle East Region and Beyond"

_microorganisms, 2021, doi:10.3390/microorganisms9020338_

Round 1
Reviewer 1 Report
This is a comprehensive review bringing together epidemiological data of malaria incidence in the Middle Eastern region. As such, it is a useful and interesting overview of the situation in the various countries in this region that gives a picture of the endogenous transmission levels and how these are affected by incoming nationals from other countries, where malaria might be more endemic.
Overall, the manuscript is well written and presents useful up to date information, however there are aspects of this manuscript that would benefit from improvement as suggested below.
- In general, the text could be better organised, following a chronological sequence of the statistics presented and a better integration of the references used, which would also avoid quite a bit of repetition in some sections.
Some specific examples are:
- The introduction relies mainly on the WHO malaria report with some references around it, but a better integration of the available literature would make the story telling more logical and easier to follow.
- The end of the first paragraph of section 3 is a bit confusing, it could be written more clearly, ie, name those countries from the middle east that are included in the WHO Eastern Mediterranean region so that it becomes clearer what prevalence is being considered where. Perhaps inclusion of a map would help (see below).
- The section about Qatar presents very interesting data but it would be good to have some numbers for the relative contributions of vivax and P. falciparum and also about the different strains and drug resistance, perhaps a little table could be included presenting these? This section talks about 0.2% of positivity rate in blood donors and about high genetic diversity of P. vivax cases and that gametocytes were detected by q-PCR that represent a threat for re-introduction of malaria. But this is rather vague, more precise statistics if available should be presented and if they are not available, this should be stated. In order to discuss a potential threat, this data needs to be known.
- In Kuwait, the same pattern is described that most of the population consists of foreign workers and the numbers from the last decade show 250 – 400 annual malaria cases, with 80% from external adults (71% vivax and 27% P. falciparum). Does that mean that 20% of cases are endogenous then?
- The section about Yemen could do with re-writing to integrate the numbers and arguments. This section also has some inconsistencies that need fixing. It starts by stating that there has been a substantial increase in malaria between 2015 and 2019, then it states that the incidence has decreased over the last 2 decades. Then it gives average numbers of deaths between 1225 and 2886 from 2000-2019, but then it quotes the number of deaths in 2018 as 57. All this data needs to be reconciled and presented in a more understandable way.
- Given that unrest and war have such an enormous impact on infectious diseases, it would be better to quote the numbers of years of conflict that some of the countries in this region have endured, and this should be more critically discussed, with the backing statistics.
- More information would be desirable about the Cyprus outbreaks, particularly as these lead to the conclusion that they represent a potential for re-emergence of malaria in Cyprus.
- Some of the numbers in the last section are confusing: if Ehtiopia accounts for 94% of all malaria cases and deaths globally, then it is not possible that Nigeria contributes 27% of all malaria cases and 23% of all malaria deaths globally and be the country with the highest malaria incidence even in the African region.
- The use of google as a source of information is not appropriate. There are several places where this is the case, like geographic location, but it would be much more precise and informative to include a map of the region and the countries contained within it and its neighbours. Furthermore, to quote www.google.com, not even a specific link, to give the census of Bahrain is not acceptable. Surely there are some much more accurate and recent governmental figures.
- Table 1 presents the statistics of malaria infection in countries where some workers in the GCC countries come from, but this does not reflect the number of cases brought into the GCC region from these countries, which would be useful and more informative to know. Perhaps adding columns to table 1 with the internal GCC statistics of approximations to the number of workers from these countries they receive and the number of cases reported amongst them. Furthermore, a table or tables summarising the data regarding the population of the countries and the prevalence of plasmodium species, the proportion of immigrants of different origin and the prevalence of malaria species in the different groups, would help understand the influx of cases and the dynamics of transmission.
- The subtitle of section 6 needs to be changed: these countries might be where some of the imported cases originate from but they are not ‘responsible’ for malaria in the GCC, which is how it sounds and gives the wrong impression.
- More detail on the statistics quoted is needed. Some aspects are not well considered at all, such as own nationals’ travel and emigration in particular, despite there being reference of many GCC nationals living abroad.
- A more in-depth discussion of the data and its implications and context in the global situation of malaria would give the manuscript more significance. For example the detection of variation in genes correlated with drug resistance and escape to detection, its distribution in the region and its implications. Cases in the migrant workforce and/or their families have been reported for decades, but there are no cases in the local population, how is this explained? In the case of Kuwait, there is the argument that it is the environmental conditions that are not favourable for the vector, so there is not transmission, is this the case for all countries? And the environmental changes brought by agriculturisation that create better conditions for mosquito breeding, what is the scale and the potential impact of this and what are the precautions that should be considered?
- The second paragraph of the conclusion section is almost entirely a repetition of the introduction. There are conclusions as to what essential measures need to be taken but there is no real discussion about the situations described; ie what do the numbers mean, are there any trends that might give indications of what to expect in the future. Mainly, an integration of the data presented and its implications into the global picture is missing. The problem of mobility of people: how to consider the health aspects of the positive social and economic impact; the consequences of displacement of people due to conflict and poverty and the lessons to be learnt from these situations on an international scale, how does it affect transmission of malaria and the spread of genetic variants that will put into jeopardy global efforts to control the disease.
Minor points:
The last sentence in page 6 could be clearer: if only two countries have reported indigenous cases, then the remaining countries have ONLY imported cases, surely.
Sentence 282, there are reported cases but how many? The numbers do follow in the text but again it is badly organised.
line 308 parasite larvae? Shouldn’t this be mosquito larvae?
It is stated that this outbreak did not lead to re-emergence of transmission but it’s not specified whether there were infected mosquitos found or not. It would also be interesting to mention the measures adopted by the ministyry of health to bring prevalence down and how was the outbreak dealt with.
Sentence 325 to 327 could do with a bit of re-writing to make it clearer
Sentence 339 – 342 talks about a declining trend but it only mentions the incidence and number of cases imported and acquired locally, which does not give information about the declining trend mentioned.
Line 372 population ARE at risk
Sentence 525 to 527 doesn’t make sense, it states Egypt was declared malaria free in 1998 after no autochthonous malaria transmission for 3 years since the year 2000.
Line 651 THE majority of malaria cases
Author Response
Responses to Reviewer 1 are attached

Reviewer 2 Report
Current Status and the Epidemiology of Malaria in the Middle East Region and Beyond
By Al-Awadhi et al.,
This review article outstand the importance to assess the current situation of malaria in the region in order to reinforce measures regarding the elimination and surveillance at the regional level. However, in my opinion as a reader, the structure the review article could be improved e.g. including some integrative Tables and/or figures, to enrich the review, make it more attractive, and probably to catch the reader's attention more readily. As the same authors indicate in the abstract (“Most of the Middle East Region countries are malaria-free as no indigenous cases of infection have been described in recent years,”), the central idea of the review could be redirect towards imported malaria in the region as the main source of malaria and challenges for surveillance. In the introduction should introduce to the problematic of the imported malaria in the region. It is suggested to integrate the “1. introduction” with “2. The global epidemiology of malaria” as both addressed similar content. Please check the apropriate content of each section, e.g. section 3, the title does not completely harmonize with the content, as the text and Table 1 combine the malaria cases and /or situation of other countries outside the middle East region, and the next section is actually a subsection as authors indicate on page 7 “For these reasons, the GCC countries will be discussed as a sub-group in this article.” I encourage the authors to prepare tables to sort out similarities and differences on malaria epidemiology in GCC countries or other countries indicated in section 5, those tables would be highly informative “at a glance” and might denote original analysis of the information. If posible integrate in a figure the population flow and the different countries as source of imported malaria and the countries affected by that.
Also, i suggest to homogenize the information presented for all malaria endemic countries, e.g. Plasmodium species, mosquito vectors, symptomatics vs asymptomatic infections, diagnosis methods, drug resistance, etc, if no data on certain aspects is available, please indicate that. The conclusions are too long, i suggest to focus on the main problematic about malaria in the studied region. Also, the last sentence could be more specific as indicate lack of knowledge, if any, and include some specific solutions (per region, subregion or country) to streighthen malaria surveillance.
Author Response
Responses to Reviewer are attached

Reviewer 3 Report
This is an interesting and quite comprehensive review. The literature cited is adequate and generally of interest.
The only concern is on the very didactic and compartmentalized structuring. It can be useful but sometimes it is too rhythmic and perhaps a comparative analysis could help. But overall a very good review.
Author Response
Responses to Reviewer are attached

Round 2
Reviewer 1 Report
The authors have made substantial efforts to clarify the confusions raised, adding a map of the region and tables presenting a summary of the data which greatly improved the manuscript. It is a sound, easy to follow detailed review of malaria in a region not widely known that presents interesting data that will undobtedly be useful to the malaria research community.
A few typos were detected that might be worth correcting:
28 …. In sub-saharan Africa (no THE)
45 parasitic infectionS
57 less than 5 yearS old or simply under 5?
167 low-wage…… in A few (several?)
200 while A few travel-related
220 – 221 Only A few breeding sites
245 among malaria parasiteS.
349 from the areA
540 during journeyS to Africa
739 have large expatriate populationS
and just a couple of comments:
414 is the number of P. falciparum cases: 112823 correct? It is in stark contrast with the number quoted above of 64233 for the same year and as these numbers come from the same reference, it is a bit strange.
I still feel the title of section 6 might be giving an impression of blame. The word responsible implies that somehow endemic countries actively spread malaria and are therefore guilty for the disease elsewhere. This is in contrast with the text that provides a clear account od the great efforts invested by most of these countries in the control and elimination of the disease with substantial success. I think it would be much more appropriate to change the wording ‘responsible for’ to ‘at the origin of’, however, this is my personal view and the final decision about the title of this section remains with the authors.
Author Response
Reviewer comments:
The authors have made substantial efforts to clarify the confusions raised, adding a map of the region and tables presenting a summary of the data which greatly improved the manuscript. It is a sound, easy to follow detailed review of malaria in a region not widely known that presents interesting data that will undoubtedly be useful to the malaria research community.
Authors response: We thank the reviewer for the positive comments. No specific comments to respond to.
Reviewer comments:
A few typos were detected that might be worth correcting:
28 …. In sub-saharan Africa (no THE)
Authors response: ‘The’ has been deleted, as suggested by the reviewer.
Reviewer comments:
45 parasitic infectionS
Authors response: The suggested change has been made.
Reviewer comments:
57 less than 5 yearS old or simply under 5?
Authors response: The suggested change has been made.
Reviewer comments:
167 low-wage…… in A few (several?)
Authors response: The suggested changes have been incorporated in the revised manuscript.
Reviewer comments:
200 while A few travel-related
Authors response: The suggested change has been made.
Reviewer comments:
220 – 221 Only A few breeding sites
Authors response: The suggested change has been made.
Reviewer comments:
245 among malaria parasiteS.
Authors response: The suggested change has been made.
Reviewer comments:
349 from the areA
Authors response: The error has been corrected, as pointed out by the reviewer.
Reviewer comments:
540 during journeyS to Africa
Authors response: The suggested changes have been incorporated in the revised manuscript.
Reviewer comments:
739 have large expatriate populationS
Authors response: The suggested change has been made.
Reviewer comments:
and just a couple of comments:
414 is the number of P. falciparum cases: 112823 correct? It is in stark contrast with the number quoted above of 64233 for the same year and as these numbers come from the same reference, it is a bit strange.
Authors response: The number of P. falciparum cases of 112823 is according to Table Annex 3 I of 2019 Global Malaria report. The number 64233 refers to microscopy-confirmed cases only. The text on lines 411 to 414 has been expanded for greater clarity.
Reviewer comments:
I still feel the title of section 6 might be giving an impression of blame. The word responsible implies that somehow endemic countries actively spread malaria and are therefore guilty for the disease elsewhere. This is in contrast with the text that provides a clear account od the great efforts invested by most of these countries in the control and elimination of the disease with substantial success. I think it would be much more appropriate to change the wording ‘responsible for’ to ‘at the origin of’, however, this is my personal view and the final decision about the title of this section remains with the authors
Authors response: The title of section 6 has been modified as suggested by the reviewer and a few minor changes have also been made in the text (Lines 631-632 and Lines 670-671) to reflect the changed title in the revised manuscript.
Reviewer 2 Report
Current Status and the Epidemiology of Malaria in the Middle East Region and Beyond
By Al-Awadhi
Thanks to the authors for revising the manuscript, and including the Tables 1 and 2. In my opinion, the review is a bit long and there is some redundancy, though it has the elements to consider for publication.
Author Response
Reviewer no. 2
Reviewer comments:
Current Status and the Epidemiology of Malaria in the Middle East Region and Beyond
By Al-Awadhi
Thanks to the authors for revising the manuscript, and including the Tables 1 and 2. In my opinion, the review is a bit long and there is some redundancy, though it has the elements to consider for publication.
Authors response: We thank the reviewer for the positive comments. No specific comments to respond to.